# Dealing with Loved Ones’ Addiction: Development of an App to Cope with Caregivers’ Stress

**DOI:** 10.3390/ijerph192315950

**Published:** 2022-11-30

**Authors:** Roberta Renati, Natale Salvatore Bonfiglio, Dolores Rollo

**Affiliations:** 1Department of Pedagogy, Psychology, Philosophy, University of Cagliari, 09123 Cagliari, Italy; 2Department of Medicine and Surgery, University of Parma, 43121 Parma, Italy

**Keywords:** dependence, addiction, caregivers, mHealth, app, usability

## Abstract

Caregivers remain the primary source of attachment, nurturing, and socialization for human beings in our current society. Family caregivers provide 11 to 100 h of care per week to their loved ones, including emotional and social support, assistance with transportation, home care, and so on. However, caregivers find the workload challenging due to fatigue, burnout, depression, anxiety, and sleep disturbances, and sometimes also from an excessive burden. Caregiver burden and stress ultimately negatively affect family members and caregivers. The caregiver is then at risk of developing deleterious physical, psychological, social, and emotional problems such as mood and anxiety disorders. Mobile health applications (mHealth applications) can be a solution to help family caregivers care for their loved ones and also for themselves. In this study, we present the development of an mHealth application for caregivers of persons with substance use and tested its usability. We used a user-centered design and intervention (UCDI) approach to develop the app by conducting a focus group with parents of individuals with addiction problems. Four key themes were identified during the focus group: (i) information section, (ii) self-care section, (iii) how-to: stress-reduction section, and (iv) chat section. The final app was developed with the software vendor and divided into several sections that were useful for managing psychological problems (such as stress or anxiety), informing about addiction and behavioral dependency problems, and helping users find a professional or services nearby. An analysis of the results of a usability test related to the app administered to a subsample of the focus group showed that the app provided ease of use, usefulness, and satisfaction.

## 1. Introduction

The family remains the primary source of attachment, nurturing, and socialization for humans in our current society [1]. Relatives and friends, as well as neighbors, can play an essential role in providing care for loved ones or those in need. They often assist others without asking for financial compensation. They provide a huge amount of support because they are involved in caregiving for a long time, often fulfilling most of the care for the patient not only physically but also psychologically, which fosters the patient’s sociality [2,3].

The average caregiving time is 11 h per week, but this is a very rough estimate. In fact, caregiving time often depends on the relationship one has with the person asking for help and the typology of illness. Caring for people with conditions such as mental dementia or Alzheimer’s or cancer can also commit the caregiver to more than 100 h per week [2].

Typically, the caregiver assists the person who needs help with almost everything (transportation, meal preparation, housekeeping, property administration, etc.), but this assistance also includes emotional and social support; for this reason, caregivers may experience a workload that leads them to develop burnout, depression, anxiety, sleep disorders, etc. [2,4]. These conditions tend to worsen because caregivers do not have the opportunity to take “breaks” because they often cannot choose whether or not to care for a loved one or decide how to manage care [2]. This leads to a “paradoxical” situation because caregivers’ stresses in turn affect the person for whom they care, generating a vicious vortex. These negative effects may lead caregivers to abuse alcohol, increase their levels of anxiety and depression, or increase the frequency of psychological or physical abuse by the caregiver [2,5,6,7].

Substance misuse disorder is a disease that can impact not only the addicted person but also individual family members or a caregivers’ quality of life, devastatingly affecting the entire family system. For example, the burden among family caregivers of patients with substance misuse disorder has been described as a feeling of distress developed due to the presence of mental disorders in the family that assign responsibilities of patient care to one of the family members for many aspects of home, personal, and family life [8,9], leading to occupational dysfunction, frequent relapses, and frequent hospitalization of patients with substance misuse disorders. In addition, it leads to caregiver stress, forced retirement, role changes, role strain/overstrain load, distancing from friends and reduction of social activities, disruption of unusual routines, and financial pressure [9,10].

As part of this “family disease,” the caregiver is at risk for developing deleterious physical, psychological, social, and emotional problems such as mood and anxiety disorders [11]. The effects of stress include worry, anger, guilt, financial and emotional strain, marital dissatisfaction and discord, diminution in the quality of life of family members, negative impacts on the growth and development of children, and physical health consequences [12]. In addition, it adversely influences the emotional climate, identity, tasks, and family relationships by placing burdens that alter caregivers’ safety [13]. In addition, family members who live with drug-addicted patients are affected by incalculable losses such as financial instability and physical, psychological, and verbal violence that reduces the quality of life and constitute a burden for both the family and the drug user [9,14].

Therefore, the impact of substance-use disorders (SUDs) on the family and individual family members merits attention [1]. Thus, understanding and addressing these issues would decrease their burdens and improve their coping skills and overall quality of life [11].

In some situations, the caregiver does not have the resources to be supported or is unable to seek help on their own [7,15]. In these cases, digital technologies can provide an easily accessible and available way to access support, often psychological or medical. One example is mobile health applications (mHealth).

mHealth, as defined by the World Health Organization (WHO), is: “medical and public health practice supported by mobile devices, such as mobile phones, patient monitoring devices, personal digital assistants, and other wireless devices.” [16]. In recent years, partly due to the conditions caused by COVID-19, which accelerated the development and use of new technologies, especially in health care, there has been an increase in the number of apps dedicated to psychological health. Most of them are dedicated to people with addiction problems [17,18,19,20], but to our knowledge, none are dedicated to caregivers.

Given the increase in smartphone use, new technology such as mHealth apps could be an effective mechanism to support caregivers [21,22]. Therefore, we aimed to develop a new mHealth app for caregivers of persons with substance use and test its usability before going to market.

## 2. Materials and Methods

### 2.1. Recruitment and Participants

We conducted a focus group study with parents of persons with addiction problems to: (i) explore how caregivers manage problems and psychological symptoms and overcome problems; (ii) understand their opinions, preferences, ideas, experiences, and needs concerning the problem due to their relatives’ caregiving; and (iii) engage them in a co-design session. Subjects referred to a service for family members of people with addictions in Pavia (Noah srl) at which they held monthly group meetings and were recruited on a voluntary basis.

The study protocol adhered to the guidelines of the ethics committee of the University of Pavia (Italy). The personal information and sociodemographic data of the participants were handled in accordance with the relevant provisions regarding the EU Regulation (2016/679-RGDP) and the “code of deontology and good conduct for the processing of personal data for statistical and scientific purposes (order of the guarantor No. 2 of 16/06/2004)” and kept confidential. Consent was provided by each participant.

A total of 15 participants aged 45 to 70 years of whom 4 were male took part in the study.

### 2.2. Procedure

The moderator initiated the focus groups by asking questions to guide the group through topics related to the previously described themes and objectives. The moderator adopted a flexible approach and was prepared to listen and follow the ideas that emerged during the discussion. The discussions were audio-recorded with the consent of the participants. Several topics were discussed during the focus group, such as ways to take care of their mental health problems (including general depression, stress, low mood, and anxiety) and ideas for creating a mental health app. All group sessions were recorded and transcribed for coding.

Focus groups were conducted at two different times with the same participants. In the first meeting, information useful to structuring the app was collected. Subsequently, a first design and conceptual idea of the app was structured by the authors. In the second meeting, useful information and suggestions were collected with respect to the first conceptual draft of the app.

A general inductive method was used to analyze the transcripts. The transcripts were read repeatedly by two of the study authors (R.R. and N.S.B.), and the text segments were coded to identify themes. As the coding framework developed, the transcripts were reanalyzed in light of the new themes that emerged. Once this stage was completed, the main themes relevant to the research question were derived. A single reviewer then analyzed the coding (D.L.), which was validated via a check by the study authors (R.R. and N.S.B.) [23].

## 3. Results

### 3.1. App Design and Functionality

Four key themes were identified: (i) the information section, (ii) the self-care section, (iii) the “how to: stress reduction” section, and (iv) the chat section. To materialize the vision and design specifications that resulted from the focus group, we worked with a software vendor. The following section describes the app and each key theme and also shows the home screen and logo of the app (Figure 1).

#### 3.1.1. Information Section

This section provides theoretical information (easy and immediate mode) on what an addiction is, specifies the differences between substance and behavioral addictions, provides specific information on what a dual diagnosis is, and provides guidance on the role the family should be given. In addition, there is a particular section on use, abuse, and dependence on adolescents (Figure 2).

For each substance, information is given on the type of substance, how it is taken, and the main signs of the manifestation of addiction (most relevant physical and behavioral effects).

There is also a section for the geolocation of services in which access and contact details are briefly specified. The geolocation is based on the user’s location and provides indications of easily accessible public and private accredited services and private specialists available within a predetermined kilometer radius.

#### 3.1.2. Self-Care Section

This section focuses on the health of the family caregiver for the person with an addiction and provides insights into why it is necessary to take care of oneself. A profile can be created with a nickname, preferred images, and completion of a questionnaire for the baseline (demographic information, health status, stressors, and coping strategies); these are assessed periodically based on a time frame set by the user.

A daily monitoring of health status perception was also added (a small checklist with a response scale or daily diary on how stressed the user feels, mood tone, or whether particular health issues have arisen). The periodic assessment allows the user to monitor the situation and progress daily, as well as the baseline, with aggregate data of the group users’ average scores (“where do you stand in relation to the average?”). The follow-ups are helpful in becoming more aware of one’s attitudes, habits, and health condition to intervene in case of “decline.” In this regard, the app provides specific messages such as “you are not taking care of yourself”, “your stress seems high today”, “devote some time, stop, ask for help”, etc. If positive, reinforcing feedback and achieved goals are given instead. In general, the app provides alerts that remind the user to take care of themselves: e.g., “did you do anything for yourself today?”, “What did you do for yourself today?”, “did you take care of yourself?”. The app provides for the inclusion of highly emotional images with impactful phrases; e.g., “you are not alone.”

#### 3.1.3. “How to: Stress Reduction” Section

This section presents games and activities aimed at reducing stress levels (Figure 3). In addition, it includes opportunities for the user to provide feedback on the effectiveness of the app and its functions (e.g., whether and how useful it is in monitoring stress levels, whether and how useful it is in using shifting/stress-reduction strategies through play or activities/tasks to perform). In addition, two types of stress-reduction techniques are presented: a hypnotic audio induction and an eye movement desensitization and reprocessing (EMDR)-like game.

#### 3.1.4. Chat Section

Focus group participants described the chat as a key element of the app that promoted the use of the app over time and provided users with concrete support from those who shared the same experience.

There are, however, numerous “dangers” and difficulties in managing the complexity of communications.

The presence of a professional to act as a moderator was imperative regardless of the possibility of reporting abuse by participants. The possibility of an initial filter was considered but was not enough because more control would be needed.

Although the idea of a group chat initially was received with great enthusiasm, when delving into the various elements that would characterize the application, the group changed its mind, fearing that the chat would become too confusing and difficult to manage. As a result, the chat would risk becoming too chaotic and risk creating a dispersive context that did not provide support via the possible intrusion of external figures who had nothing to do with the purpose of the app (third parties interested in surveys, students who wanted to get in touch with this type of user for educational purposes, or the so-called “trolls” of the Internet).

In the concluding part of the focus group, caregivers proposed the possibility of eliminating the group chat even though it was considered the primary tool for caregiver support and use of the app. Indeed, the chat could promote a more individualized relationship with a professional (1:1), with specialized staff (e.g., a moderator or tutor who could access the caregiver’s file related to physiological data, self-assessments, and feedback provided). In addition, another “more experienced” caregiver could be a moderator of the chat unless they had participated in other reliable support groups or had processed and overcome the burden related to the dependency condition of the family member with whom they could compare themselves.

Moreover, the group proposed the possibility of setting up chats at predetermined time windows for which users could book themselves. Expressly, group chats could be set up in which a small number of users and at least one moderator/specialist operator could participate. In this last case, the focus of the group would be centered on a specific topic that could be booked through the app.

We decided not to implement a chat in this prototype version of the app because of the complexity derived from the implementation of such a section due to the programming complexity, the privacy problems, and the many resources needed to control and manage such a chat.

### 3.2. Usability Testing

The mHealth app Usability Questionnaire (MAUQ) was used to assess the app’s usability [24]. The original MAUQ has 21 items on a 7-point Likert scale of agreement (from 1—strongly agree to 7—strongly disagree) with three subscales: (i) ease of use (8 items), (ii) interface and satisfaction (6 items), and (iii) usefulness (7 items).

Subjects of the focus group were asked to participate in a usability testing phase of the app prototype. At the time of the survey, the app was only available on Android devices, so only five participants were able to join because the remaining participants owned an Apple device. Subjects evaluated the app over a week of use. They averaged 63.2 years of age (6.5); the majority were male (4/5) and had an average of 16 years of education (3.9) and had used smartphone apps for 8.40 (2.3) years. Of the five participants, three were married/cohabitant, one was a widow, and one was unmarried. Moreover, two had a stable job, two were retired, and one was not employed.

Overall, the participants’ levels of usability with the prototype were below the standard cutoff, suggesting that the initial prototype had promise. This was reflected in both the overall scale and the usability subscale (see Table 1, Table 2 and Table 3). Going more deeply to the questionnaires’ responses, regarding the ease of use, the participants declared that they would probably use the app again, but the interface and the usability in social settings were not satisfying for the participants. Regarding the interface of the app, participants were eased in finding information or progress on the different sections and aspects of the app but were not so able to receive health care services. Finally, regarding the usefulness, the app seemed to ease communications with service providers but did not helped in managing the participants’ health effectively.

## 4. Discussion

In the present study, we developed an app to help caregivers of substance users manage some of their own problems. A focus group was conducted with caregivers to help design the app, and we followed the group’s suggestions. We wanted to utilize a user-centered design and intervention (UCDI) approach to develop the app [17]. Indeed, there is ample evidence that digital interventions are more likely to be successful if the target populations have an active role in their design, development, and evaluation [25,26]. In pursuit of this goal, we carefully selected the primary users to increase the likelihood of addressing user needs and expectations. Moreover, the use of techniques such as UCDI is essential when considering that most people (users or caregivers) do not engage in traditional treatment or support services due to fear of stigma or shame [27,28].

Afterward, the app was developed with the software vendor to manage psychological problems (such as stress or anxiety), to inform about dependence and behavioral addiction problems, and to help users find a professional or services in their neighborhood. The usability test results related to the app were also analyzed by administering the MAUQ to a subgroup of the caregivers’ group.

The app provided a good interface, was easy to use, and satisfied users, as reported by a subgroup of the focus group participants after compiling the MAUQ. However, some improvements must be implemented such as changes to the interface of the app and allowing the possibility of easier access to services and/or professionals.

The study aimed to develop only an initial prototype of the app. Therefore, only some of the suggestions collected in the focus groups were implemented at this early stage of developing of the app. Therefore, other suggestions such as a chat section were not implemented.

The following are additional suggestions that emerged from the focus groups and have not yet been implemented in the current version of the app. Regarding the “information” area, it should be better at providing the caregiver with a guideline for intervening in case their relative exhibits violence or manifests depressive symptoms. It should also be interesting to insert an “ask for help” button that could be directly linked to the police station or the hospital in the neighborhood.

Regarding the “self-care” area, basic physiological data should be registered to monitor stress. Moreover, based on what emerged in daily monitoring, the app should co-recommend specific exercises.

Regarding the “how to: stress reduction” area, it would be useful if the app suggests the activities and games based on the profile that emerged. It should be interesting to allow users to book an appointment with a professional at scheduled times. Moreover, data entered by the user should be accessible by a professional for treatment purposes.

A panic button could also be added and used when a user’s perceived stress levels are very high or the user is in a state of increased agitation that requires a different intervention than usual. The panic button could also help users by giving guidance on how to intervene through a few screening questions (max. 5) and address the user regarding a specific intervention related to the emergency context.

For the future direction, while also following the suggestions that emerged from the focus group, one of the most useful implementations for the app could be the recording of some physiologic responses to obtain some objective measures that can be used to assess the stress and compare it with the answers to questionnaires, and thus also be able to intervene in stress reduction. Moreover, the app could include the implementation of neurostimulation and biofeedback-device methods, both of which effectively reduce stress [29,30]. Moreover, it should be useful to implement other tips suggested by the focus group and by the MAUQ such as a chat and improvements to the interface. In this former case, a specific focus group could be used in order to follow new and interesting suggestions, or an interesting algorithm could be implemented [26].

This study had some limitations that should be overcome. First, more than one focus group should have been formed to collect more suggestions to implement and to develop the app or help the authors and software vendors to plan for the following app versions. The usability evaluation should not have involved a subgroup of the subjects involved in the focus group section and more than five subjects for generalizability and representative purposes. We intend to overcome these limitations in a successive study with the aim not only to evaluate the usability but also the feasibility of the app with a more representative group of subjects.

## 5. Conclusions

To our knowledge, there was a lack of an app specifically dedicated to caregivers of persons with behavioral addiction and substance-use problems. This was, therefore, the first attempt to develop an app in this context and to identify the salient features such an app should have. Although this work had several limitations, it should still be considered the first attempt to direct further and future research in this direction and to suggest developing an app for caregivers of persons with addiction and substance-dependence problems. In this direction, it would be helpful to design studies that involve not only users (as we did in this study) but also health professionals, designers, and developers to create apps and programs to reduce stress and help caregivers manage their difficulties.

## Figures and Tables

**Figure 1 ijerph-19-15950-f001:**
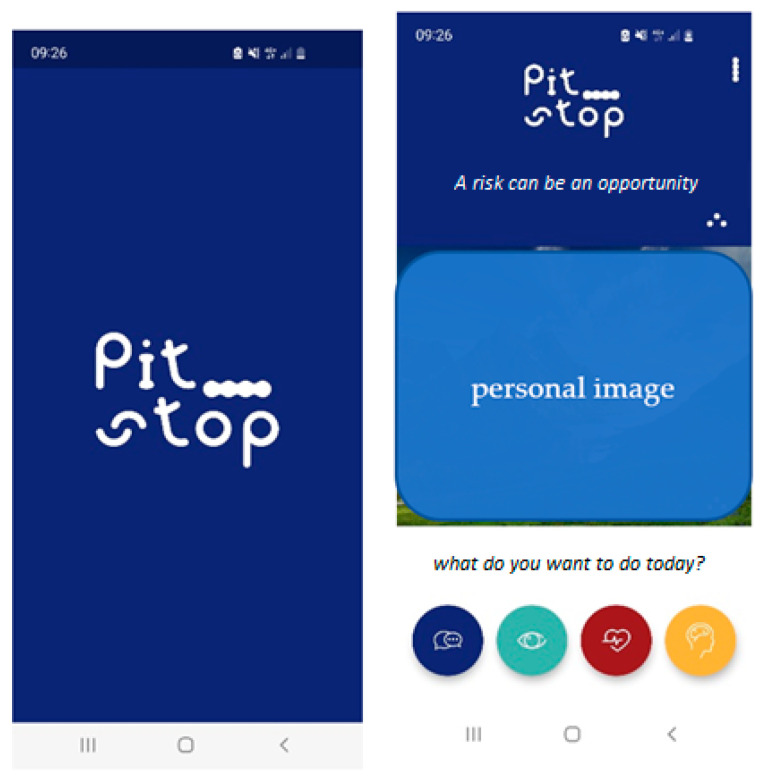
Screenshot of the home screen of the app.

**Figure 2 ijerph-19-15950-f002:**
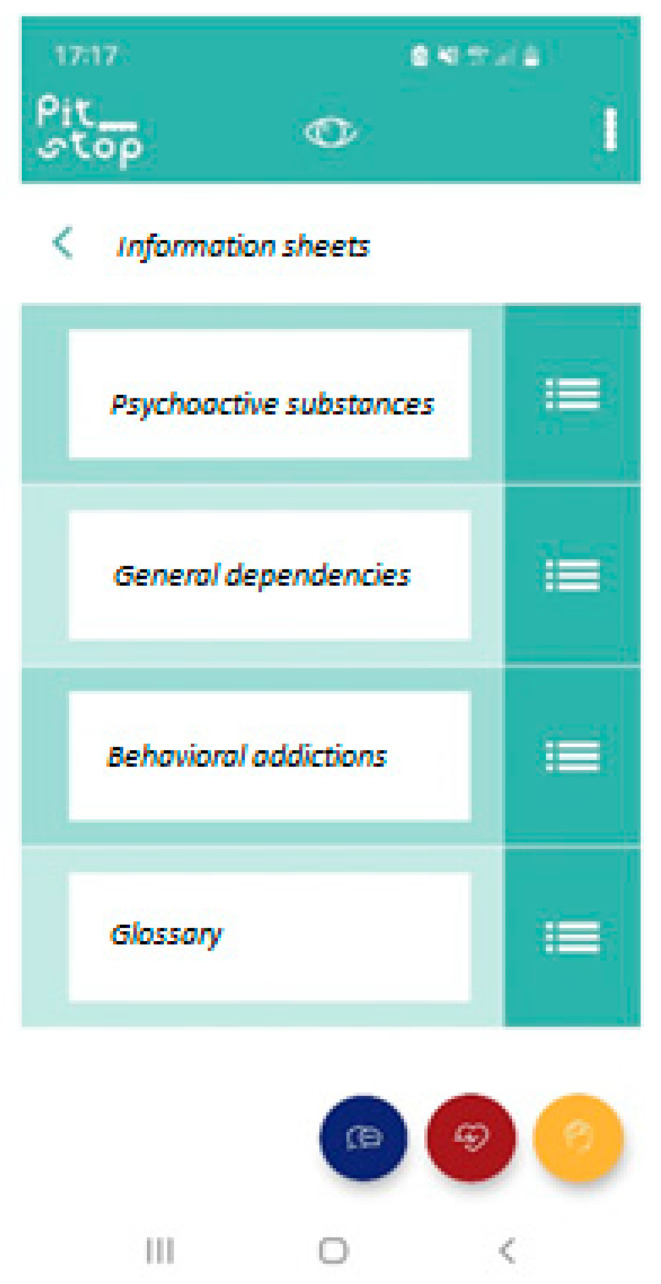
Screenshot of the information section.

**Figure 3 ijerph-19-15950-f003:**
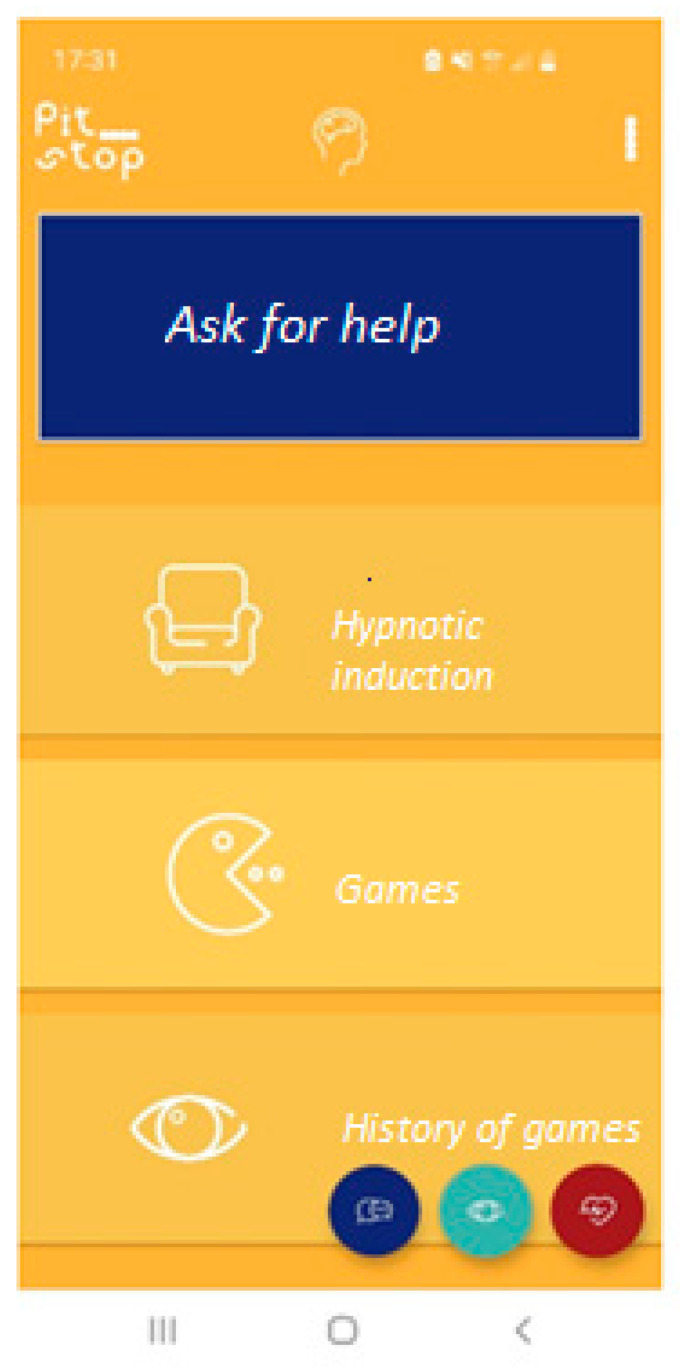
Screenshot of the games section.

**Table 1 ijerph-19-15950-t001:** Means and standard deviations in the ease of use section.

	Item Description	Mean	SD
Ease of use	*The app was easy to use.*	2.40	1.517
	*It was easy for me to learn to use the app.*	3.00	1.871
	*I like the interface of the app.*	3.20	1.789
	*The information in the app was well organized, so I could easily find the information I needed.*	2.20	1.304
	*I feel comfortable using this app in social settings.*	3.20	2.280
	*The amount of time involved in using this app has been fitting for me.*	2.40	0.894
	*I would use this app again.*	1.40	0.894
	*Overall. I am satisfied with this app.*	2.80	1.304

**Table 2 ijerph-19-15950-t002:** Means and standard deviations in the interface and satisfaction section.

	Item Description	Mean	SD
Interface and satisfaction	*Whenever I made a mistake using the app, I could recover easily and quickly.*	3.40	2.881
	*This mHealth app provided an acceptable way to receive health care services.*	3.80	3.899
	*The app adequately acknowledged and provided information to let me know the progress of my action.*	1.80	1.924
	*The navigation was consistent when moving between screens.*	2.20	1.643
	*The interface of the app allowed me to use all the functions (such as entering information, responding to reminders, viewing information) offered by the app.*	1.60	0.894
	*This app has all the functions and capabilities I expect it to have.*	2.60	3.209

**Table 3 ijerph-19-15950-t003:** Means and standard deviations in the usefulness section.

	Item Description	Mean	SD
Usefulness	*The app would be useful for my health and well-being.*	2.00	2.550
	*The app improved my access to health care services.*	3.20	3.962
	*The app helped me manage my health effectively.*	3.40	2.302
	*The app made it convenient for me to communicate with my health care provider.*	1.60	1.517
	*Using the app, I had many more opportunities to interact with my health care provider.*	2.80	2.775
	*I felt confident that any information I sent to my provider using the app would be received.*	2.20	3.271
	*I felt comfortable communicating with my health care provider using the app.*	3.20	2.280

## Data Availability

No data available.

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
