# Peer review of "Dealing with Loved Ones’ Addiction: Development of an App to Cope with Caregivers’ Stress"

_ijerph, 2022, doi:10.3390/ijerph192315950_

Round 1
Reviewer 1 Report
Dear authors,
Thank you for undertaking this study on the development of an app to cope with caregivers stress for those caring for people with addiction.
I had some feedback and comments for improvement.
1. More information on the study sample. How were the parents of people with addiction problems recruited? Was ethics obtained to do this.
2. p.3 line 89 "In addition, we conducted two focus groups on a total of 15 participant". I think the first focus group was the content covered under section 2.1 line 94-98 and then the second focus group was covering lines 102-106. Please make it clearer by saying in line 101 "the second focus group"...
3. These are now mainly grammar and consistency items now:
(i) Please be consistent and call it APP or app throughout the paper but not both. I prefer "app".
(ii) line 118 -keep it as an "information session" (as you have put in line 124). You are chopping and changing between "informative" and "information" throughout the paper.
(iii) Abstract - line 12, "However, caregivers find the workload challenging due to.." reads bettter. Also, your abstract waffled on a bit, the 1st 1/2 could have been tightened up a bit.
(iv) Line 34 "and often provide most patient care" - not sure what you mean?
(v) Line 76-78 unnecessary gap
(vi) Line 81 "Has been recently" doesn't make sense, perhaps "There have been 175 recently identified..."
(vii) Line 164 "technic" should be "techniques are presented": a hynoptic audio induction game which is similar to an EMDR..."
(viii) Line 170 Note here you use "app" after saying "APP" in first few pages
(ix) Line 172 "There are, however, numerous..." is anew paragraph.
(x) Line 176 "but more is needed" does not make sense? Do you mean more consideration of this is needed?
(xi) Line 185 to 192 is way too long for one sentence, please use a full stop somewhere. Line 185 should be "proposed". Also, please say more on why the carers didn't want the group chat?
(xii) Line 194 change "should" to "could"
(xiii) Line 207 - why only 5 parents selected? How did you decide who? How long did they use the app for?
(xiv) Line 218-222. English is weak and not clear, please improve. Line 242 should be "After the app had been developed". Line 250 "possibility for easier access" not "to". Line 254-255 doesn't make sense - please clarify what you mean. Line 256 say "information" instead of "informative". Line 264 say "could" instead of "should".
References - put them in number order (or however the journal requires).
Overall, an interesting study with potential to help the caregivers.
Author Response
I had some feedback and comments for improvement.
1. More information on the study sample. How were the parents of people with addiction problems recruited? Was ethics obtained to do this.
Thank you for your suggestion. Information about recruitment and ethic were added to the paper (see review mode tracks)
2. p.3 line 89 "In addition, we conducted two focus groups on a total of 15 participant". I think the first focus group was the content covered under section 2.1 line 94-98 and then the second focus group was covering lines 102-106. Please make it clearer by saying in line 101 "the second focus group"...
Thank you for your suggestion. A paragraph in the procedure section has been added to clarify how the focus group were conducted in two different moments. (see review mode tracks)
3. These are now mainly grammar and consistency items now:
(i) Please be consistent and call it APP or app throughout the paper but not both. I prefer "app".
(ii) line 118 -keep it as an "information session" (as you have put in line 124). You are chopping and changing between "informative" and "information" throughout the paper.
(iii) Abstract - line 12, "However, caregivers find the workload challenging due to.." reads bettter. Also, your abstract waffled on a bit, the 1st 1/2 could have been tightened up a bit.
(iv) Line 34 "and often provide most patient care" - not sure what you mean?
(v) Line 76-78 unnecessary gap
(vi) Line 81 "Has been recently" doesn't make sense, perhaps "There have been 175 recently identified..."
(vii) Line 164 "technic" should be "techniques are presented": a hynoptic audio induction game which is similar to an EMDR..."
(viii) Line 170 Note here you use "app" after saying "APP" in first few pages
(ix) Line 172 "There are, however, numerous..." is anew paragraph.
(x) Line 176 "but more is needed" does not make sense? Do you mean more consideration of this is needed?
(xi) Line 185 to 192 is way too long for one sentence, please use a full stop somewhere. Line 185 should be "proposed". Also, please say more on why the carers didn't want the group chat?
(xii) Line 194 change "should" to "could"
(xiii) Line 207 - why only 5 parents selected? How did you decide who? How long did they use the app for?
(xiv) Line 218-222. English is weak and not clear, please improve. Line 242 should be "After the app had been developed". Line 250 "possibility for easier access" not "to". Line 254-255 doesn't make sense - please clarify what you mean. Line 256 say "information" instead of "informative". Line 264 say "could" instead of "should".
Thank you for your suggestion. All the above points have been addressed and corrected (see review mode tracks)
References - put them in number order (or however the journal requires).
Thank you for your suggestion. References have been ordered (see review mode tracks)
Overall, an interesting study with potential to help the caregivers.
Thank you very much for your appreciation

Reviewer 2 Report
Thank you.
The authors should revise methodology of manuscript with more details.
Author Response
The authors should revise methodology of manuscript with more details.
Thank you for your suggestion. Methodology details have been added (see review mode tracks)
